# Are Botanical Biopesticides Safe for Bees (Hymenoptera, Apoidea)?

**DOI:** 10.3390/insects14030247

**Published:** 2023-03-02

**Authors:** Roberto Catania, Maria Augusta Pereira Lima, Michele Potrich, Fabio Sgolastra, Lucia Zappalà, Gaetana Mazzeo

**Affiliations:** 1Dipartimento di Agricoltura, Alimentazione e Ambiente, Università degli Studi di Catania, 95123 Catania, Italy; 2Departamento de Biologia Animal, Universidade Federal de Viçosa, Viçosa 36570-900, Brazil; 3Laboratório de Controle Biológico, Universidade Tecnológica Federal do Paraná—Dois Vizinhos (UTFPR-DV), Paraná 85660-000, Brazil; 4Dipartimento di Scienze e Tecnologie Agro-Alimentari, Alma Mater Studiorum Università di Bologna, 40127 Bologna, Italy

**Keywords:** ecotoxicology, pollinator, risk assessment, social bees, solitary bees, toxicity

## Abstract

**Simple Summary:**

Synthetic pesticides are among the main threatening factors for wild and managed bees. In recent decades, botanical biopesticides have been gained increasing interest and use in agriculture due to their high selectivity and short persistence in the environment. To date, however, little has been discovered or researched about the adverse effects of these substances on bees. This paper reviews studies in the literature reporting the lethal and sublethal effects of botanical biopesticides on social and solitary bees. Although botanical products are considered safer than chemical pesticides, some of them can cause lethal and several sublethal effects on bees. We suggest that more research is needed on this topic, especially increasing knowledge about certain groups of bees such as solitary bees.

**Abstract:**

The recent global decline in insect populations is of particular concern for pollinators. Wild and managed bees (Hymenoptera, Apoidea) are of primary environmental and economic importance because of their role in pollinating cultivated and wild plants, and synthetic pesticides are among the major factors contributing to their decline. Botanical biopesticides may be a viable alternative to synthetic pesticides in plant defence due to their high selectivity and short environmental persistence. In recent years, scientific progress has been made to improve the development and effectiveness of these products. However, knowledge regarding their adverse effects on the environment and non-target species is still scarce, especially when compared to that of synthetic products. Here, we summarize the studies concerning the toxicity of botanical biopesticides on the different groups of social and solitary bees. We highlight the lethal and sublethal effects of these products on bees, the lack of a uniform protocol to assess the risks of biopesticides on pollinators, and the scarcity of studies on specific groups of bees, such as the large and diverse group of solitary bees. Results show that botanical biopesticides cause lethal effects and a large number of sublethal effects on bees. However, the toxicity is limited when comparing the effects of these compounds with those of synthetic compounds.

## 1. Introduction

Bees (Hymenoptera, Apoidea) are the main group of pollinating insects, represented by about 20,000 described species in the world, with the greatest biodiversity in Mediterranean and xeric climate regions of the globe [1,2]. Due to its biological and ethological characteristics, this group provides the ecological service of pollination for spontaneous and cultivated plants. In particular, the pollination service by animals includes 87% of the world’s spontaneous flowering plants [3], and about 75% of cultivated crops [4,5]. It has been estimated that crop pollination by insects has a value ranging from USD 195 billion to ~USD 387 billion annually worldwide [6]. Despite the difficulties in estimating the economic benefits of insect pollination for wild plants, it is assumed that this service is extremely important for the maintenance of the majority of terrestrial ecosystems.

In addition to being represented by large species richness, bees also include a wide range of life history, biological, and ethological traits [7]. The majority of bees are solitary, with females that build and provision the nest and raise their offspring by themselves without cooperation with other individuals [7]. The remaining insects are represented by parasitic bees, such as cleptoparasitic and social parasite bees, and social bees. Although they represent only a small fraction of the total species, social bees (and in particular the western honeybee, *Apis mellifera* L.) have always received the greatest attention from the public, scientists, and bee regulation and conservation programs [8]. Despite the role of major pollinators historically being mainly attributed to the western honeybee, subsequent studies showed that a greater biodiversity of wild bees increases and improves the pollination service [9].

The decline of insects, and bees in particular, has been documented by various studies in recent decades [10,11,12,13,14,15,16,17,18,19,20]. The reduction in abundance and richness of bees has been documented in different parts of the world at local, regional, and country levels [21,22]. The expansion and intensification of agriculture and livestock farming, exposure to pollutants, anthropization and habitat fragmentation, fires, and climate change are the factors that most threaten the diversity and populations of bees [23,24,25].

Despite these several factors related to the worldwide decline in pollinators, the impact of synthetic pesticides on non-target beneficial arthropods, and in particular pollinating insects, has represented a primary concern for at least the last decade [26,27]. From the 1990s up until now, a large number of scientific works has highlighted the toxicity and side effects of neonicotinoids for bees [28,29,30], and which have resulted in restrictions in the use of these products in Europe [31,32]. However, other families of chemical compounds (such as carbamates, organophosphates, and pyrethroids) are well known to be dangerous for these insects [27,33,34]. Conversely, there is less information on the toxicity of insecticides of biological origin for bees [35,36,37,38,39].

Biopesticides include a wide variety of compounds of natural origin, ranging from botanical compounds such as plant secondary metabolites and essential oils (EOs), to entomopathogenic viruses, bacteria, fungi, and nematodes [40,41]. Toxins and venoms produced by arthropods such as spiders and scorpions [42], or by other invertebrates [43], are also considered to be biopesticides. The U.S. Environmental Protection Agency (EPA) categorizes biopesticides in three different groups: (I) biochemical biopesticides, (II) plant-incorporated protectants (PIPs), and (III) biocontrol organisms [44]. Although there is no formal definition of biopesticides at the European level, two different groups are recognized, namely, (I) living organisms and (II) natural products, excluding PIPs, which are not included by the regulatory authorities of most of the other countries. Here, we divide biopesticides into four different groups according to their origin: (I) botanicals (alkaloids, essential oils, limonoids, etc.), (II) microbials (virus, bacteria, fungi), (III) animals (nematodes, toxins, and venoms from invertebrates), and (IV) genetic (toxins from GM plants, and RNAi based products). Biondi et al. [35] summarized the non-target effects of spinosyns on beneficial arthropods, including pollinators, while Erler et al. [38] reported from the literature the impacts of entomopathogenic organisms on social and solitary bees. The review by Cappa et al. [37] summarized the detrimental effects of biopesticides on insect pollinators (including social and solitary bees, Lepidoptera, Diptera Syrphidae, anthophilous Coleoptera, and wasps), including the effects of different classes of microbial biopesticides, and the effects of azadirachtin among the botanical biopesticides. Furthermore, Ntalli et al. [39] summarized the effects on honeybees of botanical treatments used in beekeeping to control the *Apis mellifera* arthropod pests. Giunti et al. [45] summarized the effects of essential oil-based biopesticides on non-target organisms, reporting some information on pollinator insects.

In the present review, we analysed the impacts on bees of botanicals biopesticides used or potentially used in agriculture, summarizing the studies in the literature and reporting the lethal and sublethal effects of these products on the different groups of Apoidea Anthophila, such as social (honeybees, bumblebees, stingless bees) and solitary bees. We also reported a critical analysis on the detrimental effects of these eco-friendly products on bees, divided into different categories according to Acheuk et al. [46].

## 2. Botanical Biopesticides

Botanical pesticides have been applied for more than 150 years in Europe and North America, going back much earlier than the discovery and the wide spread use of the main classes of synthetic pesticides in the first half of the twentieth century [47]. The extensive use of synthetic pesticides, with their consequent negative effects on human and environmental health, has led to a recent and increasing demand for botanical and organic pesticides as eco-friendly alternatives to synthetic pesticides [46]. Botanical pesticides, and in general biopesticides, have a higher selectivity, cause less pest resistance, and have low environmental persistence in comparison with the synthetic compounds [46,48]. For these reasons, these products can be good candidates for modern and sustainable agriculture. Despite the growing interest of the scientific community in botanical pesticides during recent years [37], their commercial use is still restricted compared to the synthetic alternatives, particularly in developing countries [49].

The great biosynthetic ability of plants enables a wide diversity and versatility of botanical compounds, which can be divided into seven different classes: (1) alkaloids, (2) essential oils, (3) fatty acids, (4) limonoids, (5) phenolics, (6) polyketides, and (7) pyrethrins [46].

Alkaloids represent a wide and highly diverse group of chemical compounds found in several botanical species: the most important examples are anabasine from *Anabasis aphylla* L. (Amaranthaceae), nicotine from *Nicotiana* (Solanaceae) species, or zygacine from Melanthiaceae species. These compounds show high insecticidal activities at low doses, and sublethal effects such as antifeeding, deterrence, and malformations [50]. Nicotine is a non-systemic insecticide that can be used for the control of a large number of pests in a confined environment [40]. Used as a fumigant, nicotine has neurotoxic action on insects, but it also shows high toxicity for birds, aquatic organisms, bees, and humans [51]. A mixture of alkaloids can be found in sabadilla, a traditional insecticidal preparation used in Central and South America and commercially used since the 1970s [40]. Sabadilla powder is prepared from *Schoenocaulon officinale* Gray (Melanthiaceae) and contains a mixture of veratridine, cevadine, and other alkaloids, and it is effective against thrips [40]. Ryania extracts from the stem of *Ryania speciosa* Vahl (Salicaceae) contain the alkaloid ryanodine, a highly toxic bioinsecticide that has been used in the USA for the control of *Cydia pomonella* L. (Tortricidae) and *Ostrinia nubilalis* Hübner (Crambidae) [40].

Essential oils (EOs) are contained in about 17,500 aromatic plant species and can be extracted mainly by steam distillation for various industrial applications, including plant protection from pests [52]. These products have been used since ancient times and can be obtained from plants belonging to the families Asteraceae (e.g., *Artemisia* spp.), Lamiaceae (e.g., *Mentha* spp., *Origanum* spp., *Rosmarinus officinalis* L., *Salvia* spp., *Thymus* spp.), Lauraceae (e.g., *Laurus nobilis* L.), and Myrtaceae (e.g., *Eucalyptus* spp., *Myrtus communis* L.) [52,53]. EOs are produced as secondary metabolites by these plants for protection against microorganisms, insects, herbivores, and allelopathic interactions [53,54]. The major constituents of EOs are low-molecular-weight terpenoids (monoterpenes and sesquiterpenes) and phenolics [52]. EOs have been recently tested on pests with successful results, but their high volatility, poor solubility in water, and easy environmental degradation make their commercialization difficult. For these reasons, some techniques have been studied to improve their efficacy, such as encapsulation in nanoparticles (NPs) (e.g., polyethylene glycol, PEG) [55], microencapsulation in SiO_2_ [56], and the use of plant powders containing EOs [57].

Fatty acids have also been used in some commercial biopesticides where they have a stabilizing function. However, some of them can have a secondary toxic effect on insect pests, for example, conjugated linoleic acid (CLA) or pelargonic acid [46]. Furthermore, preliminary studies on the fatty amid pellitorine have shown promising results for the control of mosquitoes [58] and Coleoptera post-harvest pests [59].

Limonoids are natural compounds mainly present in plants of the Rutaceae (*Citrus* spp.) and Meliaceae (Neem tree, *Azadirachta indica* A. Juss.) families. Azadirachtin from the Neem tree is one of the most widely used and studied biopesticides [49], isolated from all the parts of the plants, in particular from seeds [40]. Considered as a safe and selective product, Azadirachtin is very effective against several groups of pests, causing acute toxicity and anti-feeding and physiological effects [60], and it can also be used as a fungicide [40]. However, several studies question its safety as regards beneficial insects [61,62,63].

Phenolics, the largest group of plant secondary metabolites, perform various essential functions, from the regulation of physiological processes to defence against herbivores [64]. Abundantly present in *Thymus* spp. (Lamiaceae) plants, thymol can be used as an effective fungicide, bactericide, and also acaricide, it being effective against *Varroa* spp. (Varroidae), an important ectoparasite of honeybees [65]. The isoflavone rotenone, extracted from the roots of some species of *Derris, Lonchocarpus*, and *Tephrosia* (Fabaceae), is a neurotoxic compound used against a wide spectrum of insects and in the control of fish populations [40]. The safety of rotenone for humans and the environment has been questioned due to its high toxicity towards mammals [47].

Another large class of plant secondary metabolites is that of polyketides, biosynthesized from acetyl-CoA. Annonins, classified as acetogenins, comprise an important group of polyketides that show a wide range of biological activities such as antimicrobial and pesticidal activities [66]. Effective against Coleoptera pests [67], annonins are extracted from the seeds of neotropical *Annona* (Annonaceae) trees [47].

Pyrethrin is one of the most marketed bioinsecticides. It is a mixture of compounds (Pyrethrins I, and Pyrethrins II) biosynthesized from *Tanacetum cinerariifolium* (Trevir.) Sch. Bip. (Asteraceae) [68]. Pyrethrin has neurotoxic action that interferes with the Na^+^/K^+^ exchange pump, causing paralysis and resulting in toxicity for several groups of pests [46]. The characteristic of pyrethrins, of being particularly labile to UV from sunlight, led in the 1970s and 1980s to the development of synthetic derivatives, the pyrethroids, which are widely used nowadays [49].

## 3. Risk Assessment of Biopesticides on Bees

Most of the regulatory risk assessment for plant protection products (PPPs) (pesticides and biopesticides) uses the western honeybee as a surrogate species for ecotoxicological testing of pollinators [69,70]. In recent years it has been realized that this approach is not enough for pollinator conservation [70,71]. Sensitivity to pesticides varies according to the bee species and other factors such as body size, level of sociality, seasonality, voltinism, floral specialization, nesting behaviour, food consumption rate, overwintering strategies, sex, and caste [27,33,72,73]. This leads to different ecological impacts from the use of pesticides. Therefore, risk assessments have recently been expanded to include other bee species such as bumblebees (*Bombus* spp.) [26,44,74,75], solitary bees (*Osmia* spp. and *Megachile rotundata* (Fabricius)) [8,26,44,76,77], and stingless bees [78]. Currently ground-nesting bees, which represent about 70% of bee species [1], have not been taken into consideration in the PPP risk assessment protocols due to their difficulty in breeding, management, and use in laboratory protocols [8]. Still, today, however, knowledge regarding the ecotoxicology of the non-*Apis* bee species is scarce and certainly needs to be increased [33,79,80].

To date, there are no specific regulations and protocols for testing the effects of biopesticides on bees. Therefore, the same protocols for the chemical pesticides developed by the Organisation for Economic Co-operation and Development (OECD) are used [37]. These protocols [74,81] include laboratory chronic and acute oral/contact toxicity tests to measure ecotoxicological parameters, such as LC_50_ and LD_50_, and to evaluate sublethal effects such as paralysis, movement alterations, and the presence of moribund specimens. However, sublethal effects caused by pesticides on bees are commonplace and still little studied, particularly regarding ecotoxicological tests about the effects of biopesticides on non-*Apis* bees [82,83,84].

In addition, there are few guidelines for the risk assessment of pesticides at field and semi-field levels with honeybees, bumblebees, and solitary bees [26,85], and no specific protocols for biopesticides. In general, the number of higher tier risk assessment studies on bees is low, both for synthetic pesticides (excluding neonicotinoids) and biopesticides.

We therefore highlight the need to (I) develop specific protocols to assess the lethal and sublethal effects of biopesticides on bees from different species; (II) increase the knowledge about species sensitivity distribution regarding chemical and biological pesticides for Apoidea, including ground-nesting solitary bees in the studies; and (III) develop new and better field and semi-field protocols both for synthetic and biopesticides.

## 4. Materials and Methods

The search for peer-reviewed English-language publications up to 2022 was conducted using Google Scholar, Scopus, and ResearchGate through the following keyword terms and their combination: “bioinsecticides”, “biopesticides”, “botanical insecticides”, “Azadirachtin”, “Essential oils”, “EOs”, “botanical extracts”, “Pyrethrins”, Pyrethrum”, “Nicotine”, AND “toxicity”, “exposure”, “effect”, AND “bee”, “social bee”, “honeybee”, “*Apis*”, “bumblebee”, “*Bombus*”, “stingless bee”, “Meliponini”, “solitary bee”, “*Osmia*”, “*Megachile*”. Additional studies from literature references were considered. In this review, papers about the toxicity on *A*. *mellifera* of botanical products used in beekeeping were not considered, as they are summarized in the recent revision of Ntalli et al. [39]. Table 1 includes and summarizes the studies and divides them following the different groups of Apoidea (social and solitary bees). Information is provided about the botanical substances tested, the category of assay (laboratory-assessed lethal (L) and/or sublethal (S) effects, field and semi-field), the type of treatment application (contact, topical, ingestion, fumigation, spray, crop spraying, crop granules, and "ingestion and topical" in the cases of tests with larvae), the target of the experiments (eggs, larvae, adults, colony, microcolony), the main effects reported by the results of the experiments, and the country in which the experiments were performed. The “Botanical substance” column in Table 1 includes the individual biopesticides tested in the different papers analysed. No papers were found in which synergistic effects between different substances were tested. Table 1 includes only article papers; however, some studies presented at conferences or symposia are discussed in the text. Most of the studies reviewed (53.7%) did not test a chemical insecticide as a positive control, and these studies are highlighted in Table 1 with a double asterisk (**). The toxicological parameters (LC_50_, LD_50_) of botanical biopesticides extrapolated from the analysed papers are reported in Table 2.

## 5. Effects of Botanical Biopesticides on Eusocial Bees

### 5.1. Honeybees

Honeybees are eusocial bees and are among the best known and most widely studied insects. Of the ten *Apis* species [86], two are managed in the world [87]. The first is the eastern honeybee, *A*. *cerana* Fabricius, managed in South and East Asia [88], and the second is the western honeybee, *A*. *mellifera*, managed in Africa and Europe for several millennia [89]. Nowadays, *A*. *mellifera* is a cosmopolitan species and is considered the most important pollinator worldwide, it being also the most important species for honey, pollen, propolis, and wax production [90]. Colony collapse disorder syndrome (CCD), observed for the first time in the US in the spring of 2007, has sparked growing interest from the public and the scientific community regarding the conservation of honeybees, and consequently other bees [7].

In the literature on the effects of botanical insecticides, both species of *Apis* previously mentioned have been studied, although *A*. *mellifera* is present in a greater number of studies. A single study [91] includes also the giant honeybee, *A*. *dorsata* Fabricius, a wild honeybee found from India to Vietnam.

Some studies showed that alkaloids can be toxic to *A*. *mellifera*. Nicotine increases the mortality, specifically of honeybees, causing a mortality from 75 to 100%, and shows a LC_50_ value of 60.15 ng/bee in contact exposure and 32.45 ng/bee in ingestion [92,93]. Sabadilla dust at its highest dosage shows 100% mortality within 48 h [94,95], and ryania dust extract at 40% caused 31% mortality 72 h after treatments [94].

Several studies have evaluated the impact of essential oils, with potential use in beekeeping for controlling *Varroa* spp. mites or other parasites, on *A*. *mellifera*, and the majority of these compounds showed low toxic effects for honeybees (reviewed by Ntalli et al. [39]). Many EOs and botanical extracts for use in crop protection also showed a lack of or low toxicity for honeybees at realistic field dosages [93,96,97,98,99]. However, several oils and extracts, widely used as biopesticides, caused lethal and sublethal effects on the larvae and adults of honeybees. Andiroba oil (*Carapa guianensis* Aublet, Meliaceae) and garlic extract (*Allium sativum* L., Amaryllidaceae) caused high larval mortality and affected the development and body mass, while citronella (*Cymbopogon* sp., Poaceae) and eucalyptus oil (*Eucalyptus* sp., Myrtaceae) showed high mortality for adult honeybees [100]. In addition, eucalyptus and garlic oil decreased honeybee speed and movement in walking tests, and all botanical treatments showed repellent effects on worker honeybees [100]. *Artemisia absinthium* L. and *Eupatorium buniifolium* Hook. ex Hook. & Arn. (Asteraceae) EOs, potentially usable against tomato pests, were tested also on *A*. *mellifera* [98] in topical tests and in the “Complete Exposure Test” described in Ruffinengo et al. [101]. Results showed a high LD_50_ in topical tests (respectively 197 and 252 μg/bee) but a low LD_50_ in the Complete Exposure Test (respectively 0.26 and 0.15 mg/cm^2^). This suggests that the use of these products may be minimally toxic for bees at the doses that can be used for the control of *Trialeurodes vaporariorum* (Westwood) (Aleyrodidae) (LD_50_: 0.08 and 0.02 mg/cm^2^, respectively) but toxic at the doses usable against *Tuta absoluta* (Meyrick) (Gelechidae) (LD_50_: 0.50 and 0.65 mg/cm^2^, respectively) [98]. EOs of *Origanum vulgare* L. and *Thymus vulgaris* L. (Lamiaceae) showed higher mortality in topical and contact tests with adult honeybees, and *O*. *vulgare* EO also reduced honeybees’ mobility during the walking bioassays [102]. Spray and ingestion treatments with extracts of *Origanum majorana* L. (Lamiaceae), *Punica granatum* L. (Lythraceae), *Echinodorus grandiflorus* (Cham. & Schltdl.) Micheli (Alismataceae), and *Matricaria chamomilla* L. (Asteraceae) reduced the survival of the honeybees, with a lower toxicity of *E*. *grandiflorus* and *M*. *chamomilla* [103]. The ingestion of *Origanum majorana* and *P*. *granatum* also reduced the mesenteric cells of the midgut of workers [103]. In addition, repellent effects on *A*. *mellifera* were observed using garlic and citronella extracts [104].

Currently, no field or semi-field studies have been conducted with EOs or their extracts to assess their effects on honeybees.

A single study evaluated the effects of fatty acid-based bioinsecticides on *A*. *mellifera* [105]. The amide pellitorine increased the mortality of larvae, newly emerged adults, and adults workers [105].

There is a considerable number of laboratory and field studies regarding the impact of Azadirachtin on honeybees (*A*. *cerana*, *A*. *dorsata*, and *A*. *mellifera*). Some of those papers reported only slight toxicity [97,106,107] or the absence of effects in field applications [108]. The first studies were conducted in the 1980s and reported high mortality and reduction in the survival of *A*. *mellifera* larvae, with morphological larval abnormalities, but no larval anti-feeding effects [109,110]. Young honeybees had malformations and were unable to hatch after the treatment of small hives with Neem seed extract spray [111]. However, these symptoms have not been observed in larger colonies, and field treatments carried out did not repel honeybees from flowers [111]. The absence of field repellence of honeybees was also observed after the treatment with a Neem seed extract of canola fields (*Brassica campestris* (L.), Brassicaceae) in Canada, despite the occurrence of repellent effects in laboratory choice tests [112]. The application of Neem oil on the cells of honeybee larvae caused high mortality at higher doses, with a higher LD_50_ than those of the other insects [113]. High mortality [100,114,115], reduction in adults’ survival [116,117,118], and larval survival and development [100,118,119] were found in laboratory tests with azadirachtin. The ingestion of azadirachtin altered the haemolymph amino acid composition [114]. Azadirachtin also impaired the flight ability [117] and walking activity [100] of honeybees. Furthermore, sublethal concentrations of azadirachtin decreased immune gene expression, and inhibited the activity of polyphenol oxidase (PPO), a midgut antioxidant enzyme of *A*. *cerana cerana* workers [115]. Field and semi-field studies highlighted the detrimental effects on honeybees after azadirachtin treatments. Although azadirachtin did not affect honeybee mortality, there was a reduction in the foraging activity and brood development of honeybees placed in tunnels with *Brassica napus* L. (Brassicaceae) in semi-field experiments [120]. Thompson et al. [121] observed a reduction in colony overwintering in azadirachtin-treated colonies but no apparent effects regarding the brood and queen development. Different formulations of azadirachtin affected the number of foraging honeybees of *A*. *cerana indica*, *A*. *dorsata*, and *A*. *mellifera* in mustard crops in India [91], as well as the number of *A*. *mellifera* forager bees in Brazilian melon fields [122].

Thymol and other phenolics such as carvacrol can be used in organic beekeeping, especially in the control of *Varroa destructor* [123]. These compounds have been tested on honeybees and their parasites by several authors, and the effects are summarized in Ntalli et al. [39]. These authors, considering them toxic to honeybees, reported however that their hazard ratios were lower than those of common synthetic alternatives. Studies on the impact of rotenone on *A*. *mellifera* have been conducted since the first half of the last century [94,124,125,126]. Rotenone was evaluated as slightly harmful to *A*. *mellifera*, after the reduction in honeybee survival in topical contact tests [97]. Repellence, high mortality of adult honeybees, reduction in body mass, and modifications in walking activity of foraging honeybees were also detected after rotenone exposure [100].

As regards polyketides, we found a single study evaluating the effects of a squamocin (annonin)-based product on *A*. *cerana* in the laboratory and field [106]. In laboratory experiments, the product was considered to be slightly to moderately toxic and selective to honeybees according to the selectivity ratio (LC_50_ of beneficial species (%)/LC_50_ of pest species (%)) (LC_50_ reported in Table 2). However, the field tests showed a significant reduction in the relative abundance and in the speed of foraging honeybees in Indian mustard crop (*Brassica juncea* L. Czern., Brassicaceae) [106].

As they were among the first botanical insecticides to be commercialized, pyrethrins have been tested on honeybees since the last century. Some studies reported no effects or low reductions in mortality [94,127,128], while others found high levels of toxicity [124,125,126] (Table 2). This is likely due to the use of different formulations and different methods of product application (by contact, fumigation, spray, topical, and ingestion). Recently, it has been shown that a pyrethrum nanopesticide decreased the longevity and caused morphological alterations in the midgut of Africanized honeybees [129].

### 5.2. Bumblebees

*Bombus* Latreille includes about 250 species with annual colonies, [1], which are abundant in the Holarctic region. Several bumblebees species in Europe and North America are threatened, and their populations are in decline [12,23,130,131,132]. There are nine species of social bumblebees managed for crop pollination [87], some of them widely used in laboratory risk assessment of synthetic pesticides. *Bombus terrestris* (L.) is the model species for ecotoxicological studies on bumblebees [26] and the only bumblebee species used in studies that have evaluated the risks of botanical insecticides (Table 1). The buff-tailed bumblebee, *B*. *terrestris*, is one of the most abundant bees in the West Palaearctic [133], and it is a widely commercialized species used for pollination of several crops [134].

There is a small number of studies on botanical insecticides regarding *B*. *terrestris*, almost all on the effects of azadirachtin.

Azadirachtin caused repellence on workers and caused high mortality in treated microcolonies of *B*. *terrestris*. Different azadirachtin concentrations also caused sublethal effects on the colonies, reducing the egg-laying, production of drones, the ovarian length, and the body mass of male offspring [61]. However, various formulations and technical powders of azadirachtin used at field-recommended doses in the laboratory did not elicit side effects on the treated colonies [135]. Colonies orally exposed to azadirachtin had a slight increase in worker and drone mortality, although the treatment was toxic for queens [136]. Other formulations of azadirachtin (nimbecidine) were highly toxic to *B*. *terrestris* in an acute oral experiment [137].

Sublethal concentrations of azadirachtin also reduced the foraging of pollen by *B*. *terrestris* in a legume field [138].

A blend of *Perilla frutescens* var. *crispa* extracts and phytoncide oil was found to be particularly toxic for *B*. *terrestris*, causing a 100% mortality after one hour in a contact experiment [139]. Sublethal doses of these botanical insecticides did not affect the walking distance and velocity of the bumblebees, but the angular velocity was significantly affected. Furthermore, the bioinsecticide reduced the levels of different genes involved in metabolism (NADH dehydrogenase 1 alpha subcomplex subunit 12, NDUFA12, cytochrome b-c1 complex subunit 9-like, UQCR10-like, ATP synthase subunit b, ATP5F1, and cytochrome b-c1 complex subunit 8, UQCRQ) at five and ten minutes after treatments [139].

To date, we are not aware of the LD_50_ or LC_50_ of botanicals biopesticides for any bumblebee species (Table 2).

### 5.3. Stingless Bees

Tribe Meliponini is a large group of bees found in the tropical and subtropical areas of the world with about 500 described species [1,140]. In Central and South America and Australia, stingless bees play important roles as pollinators and honey producers [141]. Pesticides are among the major threats for these pollinators [142,143,144,145,146], and only recently have different studies assessed the risks of pesticides and biopesticides on these pollinators [27,73,145]. Meliponini appear to be more sensitive to pesticides than *A*. *mellifera* and other pollinator species [33,147]. However, further studies are needed considering the large number of stingless bee species [27].

All studies on the impact of botanical insecticides on stingless bees were conducted in Brazil, involving a total of eight species (Table 1).

Most of the EOs tested on stingless bees resulted in low lethal effects [93,102,148,149], although other studies showed high mortality after EO treatment. However, topical contact tests with *Corymbia citriodora* EO (Myrtaceae) on *Tetragonisca angustula* Latreille resulted in 100% mortality of these pollinators [150]. EOs from *Artemisia annua* L. (Asteraceae) also increased the mortality of *T*. *angustula* [151]. The handling behaviour of *Nannotrigona testaceicornis* (Lepeletier) and *T*. *angustula* was evaluated through a videotrack system, after treatment with different EOs and extracts, which showed no effects in both species [152].

Topical applications of *Annona squamosa* L. (Annonaceae) and *Ricinus communis* L. (Euphorbiaceae) extracts reduced the survival of *Trigona spinipes* (Fabricius) [148].

The EOs of *Origanum vulgare* and *Thymus vulgaris* reduced the walking speed and the travelling distance of *Trigona hyalinata* (Lepeletier) [102].

Several studies assessed the lethal and sublethal effects of azadirachtin on stingless bees in the laboratory, while few studies have been carried out in field and semi-field conditions. Topical tests with leaf and seed extracts from *Azadirachta indica* reduced the survival of *T*. *spinipes* [148]. The ingestion of azadirachtin during larval development increased the mortality and led to the production of deformed pupae and adults of *Melipona quadrifasciata* Lepeletier, despite not delaying the development time [153]. However, in adult stingless bees, the lethal toxicity of azadirachtin seems to be lower; it caused low mortality in contact and ingestion assays with *Partamona helleri* (Friese) and *Scaptotrigona xanthotrica* Moure [154]. In this study, the flight take-off of *P*. *helleri* was affected by oral ingestion [154]. Azadirachtin did not cause mortality in *M*. *quadrifasciata* and *P*. *helleri* in contact and oral exposure experiments. However, this biopesticide, in a concentration-dependent manner, caused a significant anti-feeding effect on *P*. *helleri* and repellence in *M*. *quadrifasciata* [62]. Azadirachtin proved to be particularly toxic for *P*. *helleri* queens reared in vitro during post-embryonic development, reducing survival at the higher doses and delaying development [63]. The treatment also deformed the specimens and reduced the reproductive system area of the queens [63]. Modifications in the gene expression of esterase isoenzymes (EST) and peptides were observed in *T*. *angustula* after contact tests in the laboratory and semi-field with different formulations of azadirachtin [155].

The ingestion of azadiractin caused a reduction in the gene expression of *vitellogenin* (*Vg*) of *M*. *quadrifasciata* workers, infected and uninfected with *Escherichia coli* [156]. The same study highlighted an increase in the number of haemocytes in both infected and uninfected bees due to insecticide ingestion [156].

Flower visitation rates of *Plebeia* sp. bees were not affected by azadirachtin treatment in a Brazilian melon field [122].

## 6. Effects of Botanical Biopesticides on Solitary Bees

The great majority of bee species in the world is solitary and belongs to seven different families (Stenotritidae, Colletidae, Andrenidae, Halictidae, Melittidae, Megachilidae, and Apidae) [7]. They exhibit a great variety of size, morphological characteristics, behaviour, nesting habitats, flight ranges, phenology, and nutritional requirements [1,7]. Eight species of solitary bees, mainly cavity-nesting species belonging to the genera *Megachile* Latreille and *Osmia* Panzer (Megachilidae), are managed for the pollination of crops around the world, and another 14 are potentially usable species [87]. Among these, there are a few ground nesting bee species such as the alkali bee, *Nomia melanderi* (Cockerell) (Halictidae), which are managed in North America, or *Rhophitoides canus* (Eversmann) (Halictidae) in Eastern Europe. The status of solitary bees is not well known throughout the world. In Europe, which hosts the best-known bee fauna, the latest IUCN Red List [23] assessed 60% of the species in the “data deficient” category, and the majority of the threatened species (45 spp.) are solitary. There is a clear need to improve our knowledge of the status of solitary bees in the world and to assess the risk to them of synthetic chemicals and alternative biopesticides.

The European Food Safety Authority (EFSA) suggested including the red mason bee, *Osmia bicornis* L., and the European orchard bee, *Osmia cornuta* (Latreille), as model organisms of solitary bees in the EU pesticide risk assessment scheme [26]. However, standardised test protocols to assess acute toxicity for solitary bees are still in development. The US EPA [44] has suggested the blue orchard bee, *Osmia lignaria* Say, and the alfalfa leafcutting bee, *Megachile rotundata* (Fabricius). The latter, to date widely managed for crop pollination in North America, is a Eurasian bee accidentally introduced into the US in the 1940s.

Knowledge regarding lethal and sublethal effects of botanical compounds on solitary bees is very scarce, and the few studies carried out show non-uniform laboratory protocols, with the use of different methods of application, and different life stages (eggs, larvae, newly emerged, adults of females and males). A product based on an extract of a small tropical tree, *Quassia amara* L. (Simaroubaceae) (Tecomag^®^), was found to be particularly toxic at field doses for *Osmia cornuta* eggs and larvae, with a mortality of more than 80% in a preliminary study conducted through the application of a drop of test solution in the provision of the eggs/larvae [157].

Studies conducted in North America, with adults of *Osmia lignaria*, showed low reduction in mortality with topical and ingestion treatment of Neem oil [116]. Slightly increased mortality was also registered in adults of *Osmia cornifrons* Radoszkowski with a treatment of wintergreen oil (*Gaultheria procumbens* L., Ericaceae) as a fumigant used for the control of *Chaetodactylus krombeini* Baker (Chaetodactylidae) [158]. Another study, conducted in Canada [57], tested botanical insecticides that could potentially be used to control a natural enemy of solitary bees, such as *Pteromalus venustus* Walker (Pteromalidae), a parasitoid of the alfalfa leafcutting bees, *Megachile rotundata*. This work tested fifteen plant powders against parasitoid and adult male bees in a contact experiment, highlighting a higher bee mortality with nutmeg powders [57].

The only field study [122] was conducted in Brazilian melon (*Cucumis melo* L.) fields, investigating the visitation rates of *Halictus* spp. (Halictidae) after treatments with azadirachtin. *Halictus* Latreille is a wide genus of bees that includes a scale of different social behaviours from solitary and semi-social to social species. Since the species was not specified in Tschoeke et al. [122], we reported this in the solitary bees category. *Halictus* bees showed reduced visitation intensity after treatment with a neem-based insecticide.

As with bumblebees, the LC_50_ or LD_50_ of botanical biopesticides for solitary bee species were not calculated in the studies reviewed in the literature (Table 2).

**Table 1 insects-14-00247-t001:** Laboratory, semi-field, and field studies testing the lethal (L) and sublethal (S) effects on social and solitary bees (Hymenoptera, Apoidea) of botanical biopesticides.

Bee Species	Botanical Substance	Assay	Application	Target *	Effects	Country	Year	References
**Social species**								
**Honeybees** **(*Apis* spp.)**								
*Apis cerana cerana*	Azadirachtin	laboratory (L, S)	ingestion	adults	increase in mortality at the higher doses, anti-feeding and inhibition on the immune response	China	2022	[115] **
*Apis cerana indica*	Annonin, azadirachtin	laboratory (L) and field	topical, crop spraying	adults	increase in mortality with both compounds, reduction of the number and speed of foraging bees with annonin	India	2019	[106]
*Apis cerana indica*	Azadirachtin	field	crop spraying	adults	reduction of the number of foraging bees	India	2010	[91]
*Apis dorsata*	Azadirachtin	field	crop spraying	adults	reduction of the number of foraging bees	India	2010	[91]
*Apis mellifera*	Azadirachtin	laboratory (L)	ingestion and topical	larvae, adults	increase in mortality, larvae more susceptible than adults	Brazil	2016	[118] **
*Apis mellifera*	Sabadilla dust	laboratory (L)	contact	adults	increase in mortality	USA	1958	[95]
*Apis mellifera*	Aramite (blend of natural oils)	laboratory (L)	contact	adults	low increase in mortality	USA	1952	[96]
*Apis mellifera*	Formulations containing pyrethrins, rotenone, and pine oil, three formulations containing pyrethrins	laboratory (L, S)	spray	adults	increase in mortality and knockdown effects for all formulations	USA	1990	[126]
*Apis mellifera*	Aramite, pyrethrins, rotenone, ryania, and sabadilla dust	laboratory (L)	contact	adults	increase in mortality with sabadilla, medium and low increase in mortality with the other compounds	USA	1954	[94]
*Apis mellifera*	Pyrethrum	field	spray on cage and colony	adults, colony	low increase in mortality	USA	1979	[128]
*Apis mellifera*	*Mentha piperita*, *Origanum vulgare*, *Thymus vulgaris*, and *Zingiber officinale* EOs	laboratory (L, S)	topical, contact, ingestion	adults	increase in mortality with *O*. *vulgare*, and *T*. *vulgaris*, reduction of movements with *O*. *vulgare*	Brazil	2020	[102] **
*Apis mellifera*	Neem oil, pyroligneous extract, and rotenone	laboratory (L)	topical, ingestion	adults	reduction in survival with rotenone on topical application	Brazil	2012	[97]
*Apis mellifera*	Azadirachtin	field	crop spraying	adults	no reduction of the numbers of foraging bees	USA	2004	[108]
*Apis mellifera*	Pyrethrum extract	laboratory (L)	fumigation	adults	no effects on mortality	USA	1930	[127] **
*Apis mellifera*	Rotenone, and pyrethrum extract	laboratory (L)	spray	adults	increase in mortality	USA	1932	[124] **
*Apis mellifera*	Azadirachtin	laboratory (L, S)	ingestion	adults	high mortality, effects on haemolymph amino acid composition	Egypt	2015	[114]
*Apis mellifera*	Neem oil	laboratory (L)	ingestion and topical	larvae	reduction in survival	India	2022	[119] **
*Apis mellifera*	Neem oil	laboratory (L)	topical, ingestion	adults	reduction in survival in contact application	USA	2005	[116]
*Apis mellifera*	Pellitorine extracted from *Piper tuberculatum*	laboratory (L) and field	ingestion and topical	larvae, adults	high mortality at the highest rates	Brazil	2003	[105] **
*Apis mellifera*	Azadirachtin	laboratory (L, S)	contact, ingestion	adults	increase in mortality, no repellent effects, reduction in flight ability	Brazil	2020	[117]
*Apis mellifera*	Neem seed extract	laboratory (S) and field	ingestion, crop spraying	adults	food repellency in laboratory bioassays, however no effects on the number of the foraging bees in the field	Canada	1994	[112] **
*Apis mellifera*	Neem oil	field	ingestion and topical	larvae	increase in mortality at the higher concentration	Canada	1996	[113] **
*Apis mellifera*	century plant, citronella, garlic, parsley, rue, and tobacco extracts	laboratory (S) and field	ingestion	adults	repellent effects in laboratory and in field	Brazil	2004	[104] **
*Apis mellifera*	Pyrethrum extract, pyrethrum extract in nanoparticles	laboratory (L, S)	ingestion	adults	reduction in survival, morphological alterations in the epithelium of midgut	Brazil	2019	[129]
*Apis mellifera*	Azadirachtin	field	crop spraying	adults	reduction of the number of foraging bees	India	2010	[91]
*Apis mellifera*	*Agave americana*, *Anadenanthera colubrina*, and *Nicotiana tabacum* extracts	laboratory (L, S)	contact, ingestion	adults	increased mortality with *N*. *tabacum*, low increase in mortality with the other compounds, no effects on respiration rates or flight	Brazil	2020	[93]
*Apis mellifera*	*Echinodorus grandiflorus*, *Matricaria chamomilla*, *Origanum majorana*, and *Punica granatum* extracts	laboratory (L, S)	contact, ingestion, spray	adults	increase in mortality and reduction of the length of mesenteric cells with *O*. *majorana* and *P*. *granatum*	Brazil	2020	[103] **
*Apis mellifera*	Neem oil	laboratory (L)	contact	adults	increase in mortality	India	2017	[107]
*Apis mellifera*	Neem seed extracts	laboratory (L, S)	ingestion and topical	larvae	effects on survival of larvae, no anti-feeding effects, morphological alteration	Germany	1980	[109] **
*Apis mellifera*	Azadirachtin	laboratory (L, S)	ingestion and topical	larvae	increase in mortality, no anti-feeding effects, morphological alteration	Germany	1982	[110] **
*Apis mellifera*	Geraniol and *Cymbopogon martinii* EO	laboratory (L, S)	topical, ingestion	adults	no effects on mortality, no effects on immune response, on behaviour, and locomotion	Brazil	2018	[99]
*Apis mellifera*	Neem seed extracts	field	crop spraying	colony	effects on the hatching observed in the smaller hives, morphological alteration, non-repellent effects on treated flowers	Germany	1987	[111] **
*Apis mellifera*	Azadirachtin	semi-field	crop granules, crop spraying	colony	no effects on mortality, reduction in foraging activity and brood development with spray treatment	Czech Republic	2005	[120] **
*Apis mellifera*	Pyrethrins and rotenone	laboratory (L)	topical, ingestion	adults	increased mortality with both compounds	UK	1978	[125]
*Apis mellifera*	Azadirachtin	field	ingestion	colony	colony overwintering failure, no effects on brood and queen development	UK	2005	[121]
*Apis mellifera*	Azadirachtin	field	crop spraying	adults	effects on flower visitation rates	Brazil	2019	[122]
*Apis mellifera*	*Artemisia absinthium*, and *Eupatorium buniifolium* EOs	laboratory (L)	topical, contact	adults	no effects on mortality in the topical test, increased mortality in the contact test	Uruguay	2017	[98] **
*Apis mellifera*	Andiroba, citronella, eucalyptus, and neem oil, garlic extract, and rotenone	laboratory (L, S)	contact, ingestion and topical	larvae, adults	increase in mortality of bee larvae with andiroba, neem oil, and garlic extract, reduction of body mass of adults, reduction in walking activity and repellent effects in adult workers	Brazil	2015	[100] **
**Bumblebees** **(*Bombus* spp.)**								
*Bombus terrestris*	Azadirachtin	laboratory (L, S)	ingestion	adults, microcolony	increase in mortality, repellent effects, reduction in egg-laying, in production of drones, and in ovarian length	Belgium	2015	[61]
*Bombus terrestris*	Azadirachtin	laboratory (L)	ingestion	adults	increase in mortality	Turkey	2022	[137] **
*Bombus terrestris*	mixture of *Perilla frutescens* var. *crispa* extracts and phytoncide oil	laboratory (L, S)	contact	adults	high mortality, no effects on walking behaviour, reduction in gene expression	Republic of Korea	2018	[139] **
*Bombus terrestris*	Azadirachtin	field	ingestion	adults	reduction in pollen foraging	Estonia	2009	[138] **
**Stingless bees (Meliponini)**								
*Melipona quadrifasciata*	Azadirachtin	laboratory (L, S)	ingestion and topical	larvae	increase in mortality at higher doses, development of deformed pupae and adults	Brazil	2015	[153] **
*Melipona quadrifasciata*	Azadirachtin	laboratory (L, S)	contact, ingestion	adults	no effects on mortality, no anti-feeding effects, modifications in walking behaviour, no effects on flight and respiration rate	Brazil	2017	[62] **
*Melipona quadrifasciata*	Azadirachtin	laboratory (S)	ingestion	adults	reduction of gene expression of *vitellogenin* (*Vg*), increase of the number of haemocytes	Brazil	2021	[156] **
*Nannotrigona* aff. *testaceicornis*	*Lippia sidoides* EO, and main compounds	laboratory (L, S)	topical	adults	low increase in mortality, low reduction in locomotion ability and flight orientation, avoidance effects	Brazil	2021	[149]
*Nannotrigona testaceicornis*	Andiroba, citronella, eucalyptus, and neem oil, garlic extract, and rotenone	laboratory (S)	contact	adults	no effects on handling behaviour	Brazil	2010	[152] **
*Partamona helleri*	Azadirachtin	laboratory (L, S)	contact, ingestion	adults	no effects on mortality, anti-feeding effects, no effects on walking, flight and respiration rate	Brazil	2017	[62] **
*Partamona helleri*	Azadirachtin	laboratory (L, S)	ingestion and topical	larvae of queens	increase of mortality at the higher doses, delayed development and production of deformed queens, no effects on walking behaviour, reduction in the ovarian morphometry	Brazil	2018	[63]
*Partamona helleri*	*Agave americana*, *Anadenanthera colubrina*, and *Nicotiana tabacum* extracts	laboratory (L, S)	contact, ingestion	adults	increased mortality with *N*. *tabacum*, low increase in mortality with the other compounds, no effects on respiration rates or flight	Brazil	2020	[93]
*Partamona helleri*	Azadirachtin	laboratory (L, S)	contact, ingestion	adults	low increase in mortality, no effects on overall group activity, reduction of flight take-off of worker	Brazil	2015	[154]
*Plebeia* sp.	Azadirachtin	field	crop spraying	adults	no effects on flower visitation rates	Brazil	2019	[122]
*Scaptotrigona xanthotrica*	Azadirachtin	laboratory (L, S)	contact, ingestion	adults	low increase in mortality, no effects on overall group activity, reduction of flight take-off of worker	Brazil	2015	[154]
*Tetragonisca angustula*	Azadirachtin	laboratory (S), and semi-field	contact	adults and colony	reduction in gene expression of esterase isoenzymes, changes in protein synthesis	Brazil	2020	[155] **
*Tetragonisca angustula*	*Corymbia citriodora* EO	laboratory (L)	topical	adults	increase in mortality	Brazil	2018	[150] **
*Tetragonisca angustula*	*Artemisia annua* EO	laboratory (L)	topical	adults	increase in mortality	Brazil	2018	[151] **
*Tetragonisca angustula*	Andiroba, citronella, eucalyptus, and neem oil, garlic extract, and rotenone	laboratory (S)	contact	adults	no effects on handling behaviour	Brazil	2010	[152] **
*Trigona hyalinata*	*Mentha piperita*, *Origanum vulgare*, *Thymus vulgaris*, and *Zingiber officinale* EOs	laboratory (L, S)	topical, contact, ingestion	adults	low increase in mortality, reduction in movements with oregano and thyme EOs	Brazil	2020	[102] **
*Trigona spinipes*	*Azadiracha indica*, *Lippiasidoides*, *Sapindus saponaria*, *Anonna squamosa*, *Cymbopogon winterianum*, *Corimbia citriodora*, *Jatropha curcas*, *Ricinus communis* leaf and seed extracts	laboratory (L)	topical	adults	increase in mortality with *A*. *indica*, *A*. *squamosa*, and *R*. *communis*	Brazil	2012	[148] **
**Solitary species**								
*Halictus* sp. ***	Azadirachtin	field	crop spraying	adults	reduction of flower visitation rates	Brazil	2019	[122]
*Megachile rotundata*	Ajwain, basil, cinnamon, clove, coriander, cumin, fenugreek, fennel, ginger, nutmeg, oregano, rosemary, sage, thyme, and turmeric powders (containing EOs)	laboratory (L)	contact	adult males	increase in mortality	Canada	2020	[57] **
*Osmia cornifrons*	Wintergreen oil	laboratory (L)	topical, contact	adults	increase in mortality	USA	2009	[158] **
*Osmia cornuta*	*Quassia amara* extract	laboratory (L)	contact	eggs, larvae	increase in mortality	Italy	2003	[157]
*Osmia lignaria*	Neem oil	laboratory (L)	topical, ingestion	adults	increase in mortality	USA	2005	[116]

*: Most of the studies reviewed target adult worker bees belonging to different ages (newly emerged, foragers). The table specifies whether the target belongs to other castes (queen, male). **: Studies in which there is no positive control with chemical insecticides. ***: *Halictus* Latreille is a wide genus of bees that includes a scale of different social behaviours from solitary and semi-social to social species. Since the species was not specified in Tshoecke et al. (2019), we reported this in the solitary bees category.

**Table 2 insects-14-00247-t002:** Toxicological parameters (LC_50_, LD_50_) of botanical biopesticides extrapolated from the analysed papers. The values are reported as they were reported in the papers.

Group	Bee Species	Botanical Substance	Target	Application	Toxicological Parameters	References
HONEYBEES	*Apis cerana indica*	Annonin	adults	topical	LC_50_(%)(72 h): 1.22	[106]
		Azadirachtin	adults	topical	LC_50_(%)(72 h): 1.64	
	*Apis mellifera*	*Mentha piperita* EO	adults	contact	LC_50_(%)(24 h): 13.35	[102]
			adults	topical	LD_50_(%)(24 h): 12.58	
		*Origanum vulgare* EO	adults	contact	LC_50_(%)(24 h): 0.95	
			adults	topical	LD_50_(%)(24 h): 2.03	
		*Thymus vulgaris* EO	adults	contact	LC_50_(%)(24 h): 2.61	
			adults	topical	LD_50_(%)(24 h): 3.30	
		*Zingiber officinale* EO	adults	contact	LC_50_(%)(24 h): 22.01	
			adults	topical	LD_50_(%)(24 h): 17.98	
		Pellitorine extracted from *Piper tuberculatum*	larvae	ingestion and topical	LD_50_ (μg a.i./bee)(96 h): 0.8048	[105]
			adults	ingestion	LD_50_ (μg a.i./bee)(96 h): 0.759	
				topical	LD_50_ (μg a.i./bee)(96 h): 0.357	
		Neem oil	I instar larvae	ingestion and topical	LD_50_ (μg a.i./g)(6 d): 37	[113]
			IV instar larvae	ingestion and topical	LD_50_ (μg a.i./g)(10 d): 61	
		Nicotine extracted from *Nicotiana tabacum*	adults	contact	LC_50_ (ng/bee)(48 h): 60.15	[93]
			adults	ingestion	LC_50_ (ng/bee)(48 h): 32.45	
		β-Caryophyllene extracted from *Agave americana*	adults	contact	LC_50_ (ng/bee)(48 h): 127.4	
			adults	ingestion	LC_50_ (ng/bee)(48 h): 111.2	
		Lupeol extracted from *Anadenanthera colubrina*	adults	contact	LC_50_ (ng/bee)(48 h): 222.5	
			adults	ingestion	LC_50_ (ng/bee)(48 h): 210.1	
		*Cymbopogon martinii* EO	adults	ingestion	LD_50_ (μg/bee)(24 h): 465	[99]
			adults	topical	LD_50_ (μg/bee)(24 h): 73	
		Geraniol	adults	ingestion	LD_50_ (μg/bee)(24 h): 290	
			adults	topical	LD_50_ (μg/bee)(24 h): 43	
		Pyrethrins	adults	ingestion	LD_50_ (μg/bee): 0.29–0.13	[125]
			adults	topical	LD_50_ (μg/bee): 0.15	
		Rotenone	adults	ingestion	LD_50_ (μg/bee): >60	
			adults	topical	LD_50_ (μg/bee): >30	
		*Artemisia absinthium* EO	adults	topical	LD_50_ (μg/bee)(24 h): 252	[98]
			adults	complete exposure	LD_50_ (mg/cm^2^)(24 h): 0.15	
		*Eupatorium buniifolium* EO	adults	topical	LD_50_ (μg/bee)(24 h): 197	
			adults	complete exposure	LD_50_ (mg/cm^2^)(24 h): 0.26	
STINGLESS BEES	*Nannotrigona* aff. *testaceicornis*	*Lippia sidoides* EO	adults	topical	LD_50_ (μg/bee)(24 h): 33.7	[149]
		Thymol (compound of *Lippia sidoides* EO)	adults	topical	LD_50_ (μg/bee)(24 h): 33.6	
		ρ—cymene (compound of *Lippia sidoides* EO)	adults	topical	LD_50_ (μg/bee)(24 h): 198	
		(*E*)—caryophyllene (compound of *Lippia sidoides* EO)	adults	topical	LD_50_ (μg/bee)(24 h): 306	
	*Partamona helleri*	Nicotine extracted from *Nicotiana tabacum*	adults	contact	LC_50_ (ng/bee)(48 h): 44.32	[93]
			adults	ingestion	LC_50_ (ng/bee)(48 h): 38.76	
		β-Caryophyllene extracted from *Agave americana*	adults	contact	LC_50_ (ng/bee)(48 h): 122.2	
			adults	ingestion	LC_50_ (ng/bee)(48 h): 117.1	
		Lupeol extracted from *Anadenanthera colubrina*	adults	contact	LC_50_ (ng/bee)(48 h): 200.1	
			adults	ingestion	LC_50_ (ng/bee)(48 h): 212.2	
	*Trigona hyalinata*	*Mentha piperita* EO	adults	contact	LC_50_(%)(24 h): 21.61	[102]
			adults	topical	LD_50_(%)(24 h): 16.38	
		*Origanum vulgare* EO	adults	contact	LC_50_(%)(24 h): 7.14	
			adults	topical	LD_50_(%)(24 h): 4.57	
		*Thymus vulgaris* EO	adults	contact	LC_50_(%)(24 h): 8.29	
			adults	topical	LD_50_(%)(24 h): 6.53	
		*Zingiber officinale* EO	adults	contact	LC_50_(%)(24 h): 24.17	
			adults	topical	LD_50_(%)(24 h): 32.65	

## 7. Conclusions

Increasing awareness of the risks associated with synthetic pesticides is leading to a revaluation and increased production of studies on botanically derived products. These studies are undergoing a renaissance, especially in some countries such as India, China, and Brazil, where the number of papers on this topic has grown, particularly in recent decades [159]. Although botanicals are presented as eco-friendly alternatives with high selectivity and low persistence in the environment, the knowledge regarding their posing of risks to non-target organisms is still scarce, and studies in the literature indicate several detrimental effects on pollinators. Brazil appears to be the country from which the largest number of papers analysed came (*n* = 23), followed by the United States (*n* = 10) and India (*n* = 4). For several countries, especially in Europe, the efforts made to date on this topic are rather limited (Figure 1A). Brazil has the greatest biodiversity in the world, so many botanical biopesticides are constantly being tested [160,161,162,163]. In Brazil, beekeeping activities stand out, in addition to a vast population of native stingless bees. In this sense, efforts have been made to carry out selectivity and toxicity tests of these products on these pollinators [62,63]. In general, the toxicity of botanical biopesticides is lower than that of synthetic products. However, this review highlights how the products from different classes of botanical biopesticides can cause lethal effects and a wide variety of sublethal effects (Table 3) on different groups of bees, ranging from social to solitary species, although studies found in the literature focus on just a few model species. Indeed, the great majority of the analysed papers focused on honeybees, especially *A*. *mellifera*, while very few works focused on a few other model species, such as bumblebees, and stingless bees (Figure 1B and Figure 2). Despite neotropical stingless bees only recently being the subject of risk assessment regarding pesticides and biopesticides, there is a growing number of studies on botanical substances. On the other hand, for other groups such as solitary bees, the number of studies, and the number of species and substances tested, are still scarce (Figure 2). In the literature we found toxicological parameters (LC_50_ and LD_50_) of the different botanical pesticides only for honeybees and stingless bees, with a gap for bumblebees and solitary bees (Table 2). The toxicity of botanical biopesticides for bees varies greatly among different classes of botanicals and different formulations, and in this regard, the majority of the papers analysed focused on limonoids (azadirachtin) and essential oils (Figure 1C and Figure 2). Essential oils in general have been shown to be less toxic than other botanical products, although there are several exceptions. Alkaloid products such as sabadilla or ryania extracts cause lethal effects on honeybees and have not been tested on other bee species. Azadirachtin has proven to be one of the most studied botanical insecticides concerning bees, reporting lethal effects for several bee species, and a massive presence of sublethal effects (Table 3), this despite the detrimental effects varying significantly depending on the formulations used. The toxicity of some products for bees deserves further investigation with a focus on the sublethal effects, and an increase in field and semi-field studies, which have currently been carried out in a small proportion (Figure 2). None of the trials with alkaloids, EOs, or phenolics from the analysed papers were conducted under field or semi-field conditions (Figure 2). No synergistic effects between botanicals and between botanicals and chemicals pesticides have been investigated in bees, although this is an area of recent interest and attention [164,165,166]. Furthermore, of the total number of the studies reviewed (*n* = 54), a great proportion (53.7%, *n* = 29) do not include a chemical insecticide as the positive control in the experimental procedure. Including tests with a chemical insecticide group (control) can increase the potential for analysing and considering the data and can also facilitate understanding of the results. For this reason, it is important to highlight the need for protocols. In general, due to non-uniformly used methodologies, the results often cannot be compared. In addition, we have almost no information on their residues and persistence in bee matrices and thus the potential exposure level for bees in the field. The combination of these factors makes it complex to assess and discuss the actual safety of some of these products for bees. Therefore, the need has emerged for the development of new standardized protocols for the risk assessment of plant protection products regarding the different groups of bees. This is particularly true for stingless bees and solitary bees, there being protocols for honeybees in the literature but very few protocols for bumblebees (Table 4). New protocols are also required to evaluate the great variety of sublethal effects that could affect bees. In general, botanical biopesticides seem safer for bees than synthetic pesticides. However, some products need further evaluations, with the adoption of standardized biopesticide protocols, in order to assess the risks for different bee species, from social to solitary.

## Figures and Tables

**Figure 1 insects-14-00247-f001:**
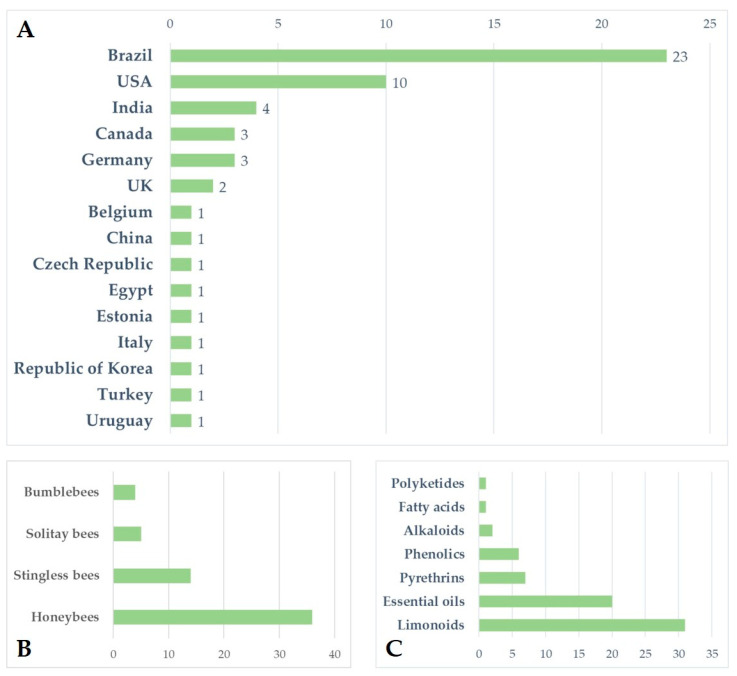
(**A**) Number of analysed papers assessing the risk of botanical biopesticides for bees by country; (**B**) number of analysed papers regarding the different groups of bees (honeybees, bumblebees, stingless bees, and solitary bees); (**C**) number of analysed papers concerning the different classes of botanical biopesticides.

**Figure 2 insects-14-00247-f002:**
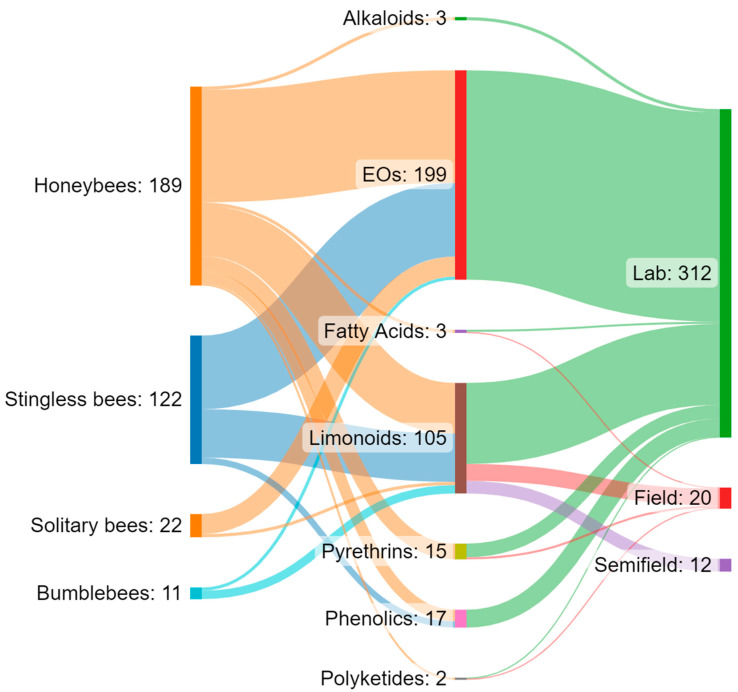
Sankey diagram showing the interaction flows between the groups of bees, different categories of botanical biopesticides, and the type of assay. The reported values refer to the number of trials (*n* = 344) found from the analysed papers (*n* = 54).

**Table 3 insects-14-00247-t003:** Classes of sublethal effects caused by botanical biopesticides reported in the reviewed studies on different groups of Hymenoptera Apoidea. In parentheses the total number of studies and the number of studies showing a significant detrimental effect.

Sublethal Effects	Honeybees	Bumblebees	Stingless Bees	Solitary Bees
Anti-feeding	Azadirachtin (3,2)		Azadirachtin (2,1)	
Brood development	Azadirachtin (3,2)		Azadirachtin (2,2)	
Foraging activity	Annonin (1,1), Azadirachtin (8,5)	Azadirachtin (1,1)		Azadirachtin (1,1)
Gene expression		EOs (1,1)	Azadirachtin (2,2)	
Immunity	Azadirachtin (1,1)			
Locomotory/flight activity	Azadirachtin (2,2), EOs (7,2)		Azadirachtin (7,3), EOs (9,4)	
Morphology and physiology	Azadirachtin (4,4), EOs (4,1), Pyrethrum (2,2)	Azadirachtin (1,1)	Azadirachtin (7,4)	
Overwintering	Azadirachtin (1,1)			
Repellence	Azadirachtin (4,1), EOs (10,8)		EOs (1,1)	
Reproduction		Azadirachtin (1,1)		

**Table 4 insects-14-00247-t004:** Overview of the research gaps in the literature regarding the effects of botanical biopesticides and bees and future directions.

Research Gaps	Future Directions
➢ Lack of specific protocols to assess lethal and sublethal effects of biopesticides on bees ➢ Use of non-uniform methodologies ➢ Lack of inclusion of a positive control (chemical pesticide) in most studies ➢ Lack of studies focusing on groups, such as bumblebees, and solitary bees ➢ Toxicological parameters (LC_50_ and LD_50_) available only for honeybees and stingless bees ➢ Few field and semi-field studies ➢ Limited efforts of European countries on this issue	Develop specific protocols to assess the lethal and sublethal effects of biopesticides on bees from different species with the inclusion of a positive control Increase the knowledge of species sensitivity distribution regarding chemical and biological pesticides for Apoidea, including in the studies the ground-nesting solitary bees Develop new and better field and semi-field protocols for both synthetic and biopesticides

## Data Availability

The data presented in this study are available upon request from the corresponding author.

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
