# Peer review of "Are Botanical Biopesticides Safe for Bees (Hymenoptera, Apoidea)?"

_insects, 2023, doi:10.3390/insects14030247_

Round 1
Reviewer 1 Report
The Review “Are Botanical Biopesticides safe to bees (Hymenoptera, Apoidea)?” gives an overview of the use and the toxicity of botanical biopesticides on different groups of social and solitary bees. The article is of particular interest as nowadays it is important to find and develop solutions which are less impacting on the environment and on animal health.
I would like to underline that line numbers are missing from line 248 onwards, which it makes it difficult to properly comment by referring to specific parts of the text.
In general, I think that the presented review gives a good overview about the topic, however I would like to suggest to better specify the toxic effect of the substances, where this is referred to further reading in literature.
Also, I suggest moderate English changes as not always there is a verb and subject-verb agreement, as an example.
I would ask to resubmit the MS indicating the lines for further revisions
Author Response
Dear Reviewer,
Thanks for your important suggestions, below is our point-by-point response.
The Review “Are Botanical Biopesticides safe to bees (Hymenoptera, Apoidea)?” gives an overview of the use and the toxicity of botanical biopesticides on different groups of social and solitary bees. The article is of particular interest as nowadays it is important to find and develop solutions which are less impacting on the environment and on animal health.
I would like to underline that line numbers are missing from line 248 onwards, which it makes it difficult to properly comment by referring to specific parts of the text.
The file we submitted to the journal had the tables at the end of the paper and line numbers throughout the article. For layout reasons the tables were moved after the Materials and Methods section, consequently the line numbering disappeared from the tables onward. The new version we submit presents the line numbers on the entire work.
In general, I think that the presented review gives a good overview about the topic, however I would like to suggest to better specify the toxic effect of the substances, where this is referred to further reading in literature.
Following the reviewer's suggestion, the toxic effects of some substances have been better specified, particularly in lines 276-280, and 347-348.
Also, I suggest moderate English changes as not always there is a verb and subject-verb agreement, as an example.
To improve the readability of the article, a language revision by native speakers was carried out.
I would ask to resubmit the MS indicating the lines for further revisions
Done.
Reviewer 2 Report
Dear colleagues! The presented work is original. The authors provided a literary review of studies on the safety of Botanical Biopesticides for bees. There is competent and persuasive introduction. In chapters 2-4, the authors described in detail the varieties of Botanical biopesticides, Risk assessment of bio pesticides onbees, Materials and methods. Quoting 166 literature sources does not raise doubts about the relevance and reliability of the material used. The authors cited their own research. The authors also described in detail the analytical results of the work. The conclusions of the research are beyond doubt. For a small improvement of the manuscript I ask the authors to make the following additions: 1) Statistical research methods that can be applied even for such works are not used here at all. Even according to Table 1, a cluster analysis can be performed. 2) In Table 1, you can provide information about the years of the study. 3) In addition to the tables, you can also bring graphic images.Author Response
Dear Reviewer,
Thanks for your important suggestions, below is our point-by-point response.
Dear colleagues! The presented work is original. The authors provided a literary review of studies on the safety of Botanical Biopesticides for bees. There is competent and persuasive introduction. In chapters 2-4, the authors described in detail the varieties of Botanical biopesticides, Risk assessment of bio pesticides onbees, Materials and methods. Quoting 166 literature sources does not raise doubts about the relevance and reliability of the material used. The authors cited their own research. The authors also described in detail the analytical results of the work. The conclusions of the research are beyond doubt.
For a small improvement of the manuscript I ask the authors to make the following additions:
1) Statistical research methods that can be applied even for such works are not used here at all. Even according to Table 1, a cluster analysis can be performed.
We considered that cluster analysis was not the most appropriate analysis for this work. As an alternative, we produced a Sankey diagram based on the data in Table 1.
2) In Table 1, you can provide information about the years of the study.
Column with the year of the publication has been added.
3) In addition to the tables, you can also bring graphic images.
To enhance the graphic impact of the article, we have prepared a graphical abstract and a Sankey diagram.
Reviewer 3 Report
- This study provides a descriptive overview of studies examining botanical substances on bee health. The tables and figures provided were generally informative and clear. This review does, with respect, recommend careful editing of the grammar and word selection in the paper to increase readability. Below are a few mostly minor suggested changes.
L228: Change “follow” to “following”
- Consider moving Tables 1 and 2 after Section 6, describing the results of the source studies would help readers better navigate and contextualize the tables
- Table 1 lists studies that have examined both individual botanical substances and a number of substances simultaneously, but it is not clear if synergistic effects were examined in those studies, perhaps consider distinguishing synergistic effects
- Section 5.1: In the first sentence of the fourth paragraph change “which” to “with”
- Varroa should be capitalized and italicized as it is a genus name
- Section 7: In the second sentence, the wording makes it sound as if botanicals are sentient, consider adjusting the wording from “botanicals present themselves…” to “botanicals are presented…”
- For Table 3, the Sublethal Effects column could be a bit more descriptive, for instance indicating directionality of some of the effects (did botanical substances increase or decrease foraging activity, etc.)
- Table 4 seems unnecessary as it is redundant with the Conclusion section and some details provided in the Introduction.
- This reviewer recommends that the authors describe a few specific components of the protocol(s) they envision being implemented for better pollinator protection
Author Response
Dear Reviewer,
Thanks for your important suggestions, below is our point-by-point response.
This study provides a descriptive overview of studies examining botanical substances on bee health. The tables and figures provided were generally informative and clear. This review does, with respect, recommend careful editing of the grammar and word selection in the paper to increase readability.
To improve the readability of the article, a language revision by native speakers was carried out.
Below are a few mostly minor suggested changes.
L228: Change “follow” to “following”
Done
- Consider moving Tables 1 and 2 after Section 6, describing the results of the source studies would help readers better navigate and contextualize the tables
Done
- Table 1 lists studies that have examined both individual botanical substances and a number of substances simultaneously, but it is not clear if synergistic effects were examined in those studies, perhaps consider distinguishing synergistic effects
Line 546 in the Conclusions section, states that no studies were found that investigated synergistic effects between botanicals or between botanicals and chemical pesticides. However, to make this clearer, the following sentence has been added in the Materials and Methods section: “The "Botanical substance" column in Table 1 includes the individual biopesticides tested in the different papers analysed. No papers were found in which synergistic effects between different substances were tested”.
- Section 5.1: In the first sentence of the fourth paragraph change “which” to “with”
Done
- Varroa should be capitalized and italicized as it is a genus name
The term varroa mites referred to the common name of the species, however we believe it is more accurate to follow the reviewer's comment and refer to the genus name.
- Section 7: In the second sentence, the wording makes it sound as if botanicals are sentient, consider adjusting the wording from “botanicals present themselves…” to “botanicals are presented…”
Done
- For Table 3, the Sublethal Effects column could be a bit more descriptive, for instance indicating directionality of some of the effects (did botanical substances increase or decrease foraging activity, etc.)
To make this clearer, the table caption has been modified as follows: “… In parentheses the total number of studies and the number of studies showing a significant detrimental effect.”
- Table 4 seems unnecessary as it is redundant with the Conclusion section and some details provided in the Introduction.
We believe that Table 4 can summarize some important highlights of the article, making them stand out to the reader's eye. Some changes were also made as a result of the reviewer's next comment.
- This reviewer recommends that the authors describe a few specific components of the protocol(s) they envision being implemented for better pollinator protection
Specific components of future protocols are discussed throughout the Conclusions section, and it could be redundant to repeat them again. In addition, Table 4 includes in the Future Directions column suggestion for the improvement of the current protocols.
The development of new protocols for pollinator risk assessments is part of the authors’ research aims and they can be included and discussed more in future works.
Round 2
Reviewer 1 Report
The MS has been improved by the Authors, however I would still point out some minor adjustments to the MS as in some lines it is not easily understood what the authors would like to discuss.
Line 114: double “as”
Lines 132-134: “ a mixture of alkaloids can be found in Sabadilla….”
Line 207: full stop is missing
Line 269: I would suggest to change the word “domesticated” as we cannot really talk about “domestication” in honey bees
Line 276: Varroa destructor or Varroa spp.
Line 286 throughout the text: As it could create some confusion, I would suggest to specify Authors are talking about are honey bees, and leave the generic “bee/bees” word when there is no specificity, also to better underli
ne the important point previously stated in lines 220-222.
Line 295: use the short EOs for essential oils
Line 355: “As they were among….. Pyrethrins”
Line 372: "is one of the most abundant bees"
Line 377: "caused high mortality in treated microcolonies..."
Lines 397-398: "To date we are not aware of...."
Lines 438-439: please rephrase
Lines 483-484: please rephrase
Lines 512-513: please rephrase
Author Response
Dear Reviewer,
Thank for your further revisions and comments.
Following is our point-by-point response:
The MS has been improved by the Authors, however I would still point out some minor adjustments to the MS as in some lines it is not easily understood what the authors would like to discuss.
Line 114: double “as”
the second "as" deleted
Lines 132-134: “ a mixture of alkaloids can be found in Sabadilla….”
modified
Line 207: full stop is missing
full stop added
Line 269: I would suggest to change the word “domesticated” as we cannot really talk about “domestication” in honey bees
modified in: "includes also the giant honeybee A. dorsata Fabricius, a wild honeybee found from India to Vietnam." removing the last sentence
Line 276: Varroa destructor or Varroa spp.
modified in "Varroa spp."
Line 286 throughout the text: As it could create some confusion, I would suggest to specify Authors are talking about are honey bees, and leave the generic “bee/bees” word when there is no specificity, also to better underline the important point previously stated in lines 220-222.
the words bee/bees have been changed to honeybee/honeybees
Line 295: use the short EOs for essential oils
done
Line 355: “As they were among….. Pyrethrins”
done
Line 372: "is one of the most abundant bees"
done
Line 377: "caused high mortality in treated microcolonies..."
done
Lines 397-398: "To date we are not aware of...."
done
Lines 438-439: please rephrase
modified in: "However, in adults stingless bees the lethal toxicity of Azadirachtin seems to be lower, it caused low mortality in contact and ingestion assays with Partamona helleri (Friese) and Scaptotrigona xanthotrica Moure"
Lines 483-484: please rephrase
modified in: "Slight increased mortality was also registered in adults of Osmia cornifrons Radoszkowski with a treatment of Wintergreen oil (Gaultheria procumbens L., Ericaceae) as a fumigant used for the control of Chaetodactylus krombeini Baker "
Lines 512-513: please rephrase
modified in: "In Brazil beekeeping activities stand out, in addition to a vast population of native stingless bees"